# Nanosystems for Brain Targeting of Antipsychotic Drugs: An Update on the Most Promising Nanocarriers for Increased Bioavailability and Therapeutic Efficacy

**DOI:** 10.3390/pharmaceutics15020678

**Published:** 2023-02-17

**Authors:** Maria Daniela Ferreira, Joana Duarte, Francisco Veiga, Ana Cláudia Paiva-Santos, Patrícia C. Pires

**Affiliations:** 1Department of Pharmaceutical Technology, Faculty of Pharmacy of the University of Coimbra, University of Coimbra, 3000-548 Coimbra, Portugal; 2REQUIMTE/LAQV, Group of Pharmaceutical Technology, Faculty of Pharmacy of the University of Coimbra, University of Coimbra, 3000-548 Coimbra, Portugal; 3Health Sciences Research Centre (CICS-UBI), University of Beira Interior, Av. Infante D. Henrique, 6200-506 Covilhã, Portugal

**Keywords:** antipsychotics, bipolar disorder, brain targeting, intranasal delivery, nanoparticles, nanocarriers, nanosystems, psychotic disorders, schizophrenia

## Abstract

Orally administered antipsychotic drugs are the first-line treatment for psychotic disorders, such as schizophrenia and bipolar disorder. Nevertheless, adverse drug reactions jeopardize clinical outcomes, resulting in patient non-compliance. The design formulation strategies for enhancing brain drug delivery has been a major challenge, mainly due to the restrictive properties of the blood–brain barrier. However, recent pharmacokinetic and pharmacodynamic in vivo assays confirmed the advantage of the intranasal route when compared to oral and intravenous administration, as it allows direct nose-to-brain drug transport via neuronal pathways, reducing systemic side effects and maximizing therapeutic outcomes. In addition, the incorporation of antipsychotic drugs into nanosystems such as polymeric nanoparticles, polymeric mixed micelles, solid lipid nanoparticles, nanostructured lipid carriers, nanoemulsions, nanoemulgels, nanosuspensions, niosomes and spanlastics, has proven to be quite promising. The developed nanosystems, having a small and homogeneous particle size (ideal for nose-to-brain delivery), high encapsulation efficiency and good stability, resulted in improved brain bioavailability and therapeutic-like effects in animal models. Hence, although it is essential to continue research in this field, the intranasal delivery of nanosystems for the treatment of schizophrenia, bipolar disorder and other related disorders has proven to be quite promising, opening a path for future therapies with higher efficacy.

## 1. Introduction

### 1.1. Schizophrenia and Other Schizoaffective Diseases: Current Treatments and Challenges

Psychotic disorders are among the most impactful psychiatric illnesses, having a great influence on the lives of patients and leading to high morbidity and mortality rates. Among them, schizophrenia and bipolar disorder are the most prevalent, affecting around 24 and 40 million people worldwide over the last decade, respectively [1,2,3,4,5]. Schizophrenia is a chronic disorder, most commonly beginning in late adolescence and early adulthood, characterized by the presence of positive symptoms, such as hallucinations, delusions and significant and intense changes in thought and behavior (such as self or hetero-aggressiveness, agitation and bizarre attitudes), negative symptoms, such as dementia, impairment of interpersonal relationships, social isolation, apathy, alogia, anhedonia and lack of motivation and initiative to perform ordinary everyday activities, and cognitive symptoms, such as marked deficits in verbal and working memory, vigilance and attention [6,7,8,9,10]. In turn, bipolar disorder is also a chronic mental illness, typically characterized by either recurrent depressive episodes, including feelings of deep sadness and energy loss, or episodes of mania/hypomania, including periods during which people feel overly exhilarated, happy, irritable and/or energetic, with a reduced need to sleep [11,12,13].

The first-line treatment for both schizophrenia and bipolar disorder is the prescription of antipsychotic drugs. Most of these drugs will act as antagonists or partial agonists of dopaminergic receptors, especially the D2 subtype, in the mesolimbic pathway of the cerebral cortex [14,15,16,17,18]. However, antipsychotic drugs also exhibit a variety of affinities towards other receptors, such as serotoninergic, cholinergic, adrenergic and histaminergic receptors, which leads to each antipsychotic drug molecule having a unique action profile, but also several systemic adverse effects. The most-used antipsychotic drugs include “typical” or “first-generation” antipsychotics, such as haloperidol, chlorpromazine, fluphenazine, perphenazine or prochlorperazine, “atypical” or “second-generation” antipsychotics, such as quetiapine, risperidone, olanzapine, asenapine, lurasidone, zotepine, amisulpride or clozapine, and “third-generation” antipsychotics, such as aripiprazole, cariprazine or brexpiprazole [19,20,21,22,23].

In what concerns the main differences between these “generations”, typically, first-generation antipsychotics were first discovered around the 1950s and act mainly as dopamine D2 antagonists. Nevertheless, many patients lacked effective therapeutic responses to these drugs, and due to lack of selectivity, they led to a series of severe adverse reactions, such as sedation, weight gain, parkinsonism, tardive dyskinesia, delirium and memory deficits. Additionally, first-generation antipsychotic drugs were primarily effective in treating positive symptoms, leaving negative symptoms untreated and sometimes even making them worse. Hence, new molecules were developed, the so-called second-generation, which act mostly on dopamine D2 and/or serotonin 5-HT2 receptors. These drugs have higher therapeutic efficacy and cause fewer systemic side effects. Additionally, in addition to efficacy against positive symptoms, these drugs are effective against some negative symptoms or at least do not tend to worsen them, which is a significant improvement. Third-generation molecules, developed more recently, have added even higher selectivity, leading to fewer side effects and overall higher efficacy for both positive and negative symptoms of schizophrenia [24,25,26].

Antipsychotic drugs are mainly commercialized as oral pharmaceuticals, but there are also some parenteral formulations available on the market for specific situations. Hence, in cases where patients do not adhere to oral therapeutic regimens or when this administration route is compromised (in the case of vomiting, nausea, hypersalivation, swallowing difficulties, etc.), the administration of antipsychotic therapy is usually performed via the intramuscular route in the form of an extended-release injectable preparation [27,28,29,30].

Nevertheless, there are numerous disadvantages associated with antipsychotic drug administration, such as the occurrence of frequent, severe and sometimes fatal adverse effects, leading to decreased quality of life and life expectancy for the patient, thus reducing adherence to therapy and, consequently, disease control. Clinically significant antipsychotic adverse effects include the following: extrapyramidal symptoms, such as akathisia, Parkinsonism, catalepsy and tardive dyskinesia; anticholinergic effects, such as dry eyes and mouth, constipation, urinary retention, cognitive impairment and memory deficits; hematological effects, such as neutropenia, leucopenia and agranulocytosis; neuroleptic malignant syndrome, which comprises symptoms such as altered mental status, stiffness, hyperthermia, autonomic overactivity, excessive sweating and urinary incontinence; metabolic syndrome, comprising symptoms such as weight gain, hyperglycemia, diabetes mellitus and altered lipid metabolism; cardiovascular effects, such as QT interval prolongation and orthostatic hypotension; endocrine effects, such as hyperprolactinemia; and respiratory diseases, such as asthma and chronic obstructive pulmonary disease [31,32,33,34].

Moreover, the oral route of administration presents several disadvantages, such as rapid drug elimination, extensive pre-systemic gastrointestinal and hepatic drug metabolism, drug–drug and food–drug interactions, extensive drug binding to plasma proteins and extensive extravascular distribution, particularly in lipid-rich tissues (such as adipose, hepatic, pancreatic and renal tissues), which means that only a small part of the drug will reach the brain and have the intended therapeutic action, requiring repeated drug administration. On the other hand, the intramuscular route of administration is invasive, leading to pain or discomfort at the injection site, with the risk of developing local inflammation or infection, and requires more time for the steady state to be achieved, which leads to the persistence of the adverse effects for a longer time period (even after the treatment is stopped). Moreover, the intramuscular administration of some antipsychotic drugs requires the concomitant administration of an oral pharmaceutic form for the first few weeks to ensure that therapeutic plasma concentrations are reached and that there is good patient tolerability to the drug [35,36,37,38]. Thus, given all the drawbacks associated with antipsychotic drug administration and the commonly used administration routes, the development of new strategies for safer and more effective therapies is necessary.

The issues related to the disadvantages of the most commonly used administration routes can be surpassed by resorting to intranasal (IN) administration. For many years IN drug administration was solely used for the local treatment of nasal cavity diseases and symptoms. However, in recent years, it has been recognized as a promising route for the delivery of drugs to the Central Nervous System, as it allows brain drug delivery through three different pathways: the indirect pathway, in which the drug is absorbed into the systemic circulation and then crosses the blood–brain barrier (BBB) in order to reach the brain; or the direct pathways, which allow the drug to reach the brain directly through neuronal transport, via olfactory nerve pathway or trigeminal nerve pathway, hence avoiding the passaging of the BBB (Figure 1) [39,40,41,42,43].

Issues related to the drug molecules, such as low solubility, low permeation, susceptibility to chemical and metabolic degradation, untargeted delivery, and, consequently, low bioavailability, which are not easily solved by using conventional formulations, can be solved by incorporating them into nanosystems [44,45,46,47].

### 1.2. Nanosystems

Nanosystems are structures of less than 1 µm in size that have numerous advantages that make them ideal vehicles for drug delivery, such as the following: high chemical and biological stability, as they protect the drugs from chemical and metabolic degradation; being constituted of biocompatible, biodegradable and non-toxic excipients, leading to reduced toxicity and immunogenicity; a small particle size, allowing them to overcome biological barriers, such as the BBB, more easily; increasing the solubility of poorly soluble drugs; having a controlled drug release capacity, hence being able to ensure constant plasma drug concentrations with minimal fluctuations over a long period of time, thus decreasing the required drug dose and associated side effects; and the ability to be administered by any route of administration as long as their characteristics are adapted to the requirements of each route. All of these advantages make them promising formulations, especially when compared to conventional liquid or semisolid preparations (such as solutions, suspensions or gels). This is applicable to all administration routes but is especially important in the case of intranasal delivery since the drug will not only be protected from the degradation caused by nasal enzymes but will also enhance its permeation, consequently affecting brain transport [48,49,50,51,52].

There are several types of nanosystems that consist of different materials, such as natural and synthetic polymers, lipids, phospholipids, organometallic compounds, etc. In the present review, only those that have been used for the brain delivery of antipsychotic drugs will be discussed, which are as follows: polymeric micelles, polymeric nanoparticles, solid lipid nanoparticles, nanostructured lipid carriers, nanoemulsions, nanoemulgels, liposomes, niosomes, spanlastics and nanosuspensions (Figure 2).

Polymeric micelles (PMs) are nanosystems with sizes ranging from 20 to 200 nm obtained from the spontaneous assembly of amphiphilic copolymers in aqueous media when present above a certain concentration, which is known as the critical micelle concentration. Given this composition, PMs have a hydrophobic inner core and a hydrophilic outer layer. To optimize the properties and overcome the disadvantages of simple PMs, such as disaggregation/dissociation upon dissolution, two or more different amphiphilic copolymers can be combined to create what is commonly called mixed polymeric micelles (MPMs). Overall, MPMs exhibit higher stability and encapsulation efficiency (EE) than simple PMs [53,54,55,56,57].

Polymeric nanoparticles (NPs) have a size between 10 and 1000 nm, and depending on the preparation method, two types of structure can be produced with different compositions and structural organization: nanocapsules or nanospheres. Nanocapsules are reservoir systems consisting of a liquid core of a lipidic nature surrounded by a thin polymeric membrane. In these nanosystems, the drug is dissolved inside the core and/or included or adsorbed to the membrane. In contrast, nanospheres do not have a differentiated core, being instead formed by a dense polymeric matrix in which the drug is uniformly dispersed [58,59,60,61,62].

Solid lipid nanoparticles (SLNs) range in size from 50 to 1000 nm and are composed of solid lipid components, such as mono-, di- and triglycerides, fatty acids or waxes, dispersed in an aqueous solution with surfactants that stabilize the system. However, these systems have some limitations, such as insufficient EE and the loss of the drug due to lipid transition to the crystalline phase during storage and are easily eliminated by the endothelial–reticulum system [63,64,65,66]. Hence, to overcome the disadvantages associated with SLNs, a new version of lipid nanocarriers was created, nanostructured lipid carriers (NLCs). These nanosystems consist of liquid lipids included in an unstructured solid lipid matrix, which makes the preparations more stable, with a lower tendency to crystallize, resulting in a higher EE [67,68,69,70].

Nanoemulsions are liquid-in-liquid colloidal dispersions with an average droplet diameter ranging from 10 to 200 nm. They are usually either biphasic, being oil-in-water or water-in-oil in nature, or triphasic, being water-in-oil-in-water or oil-in-water-in-oil dispersions. Surfactants and cosurfactants are added to these formulations in order to reduce their thermodynamic instability, interfacial tension and, in turn, droplet coalescence, with the amount and type being important factors in the formation of a stable nanoemulsion [71,72,73,74]. Additionally, under specific circumstances, there might be a need to increase the viscosity of these nanosystems, for example, in intranasal delivery, in order to allow for a longer residence time of the formulation in the nasal cavity, consequently leading to increased drug absorption. In these cases, a nanoemulgel can be formed by adding polymers, such as carbomers, poloxamer 407 and xanthan gum, to the nanoemulsion’s aqueous phase. At specific concentrations, these polymers will transform the aqueous phase into an in situ gel, which will be formed when the formulation is subjected to specific conditions, such as pH, temperature and the presence of mono or divalent cations, respectively [75,76,77,78].

Liposomes are spherical colloidal vesicles composed of one or more amphiphilic phospholipid bilayers delimiting an aqueous inner core. Their size ranges from 20 to 1000 nm, depending on the number of bilayers. These nanosystems have unique structures that make them capable of internalizing drugs with different solubilities: hydrophilic drugs in the aqueous inner core, hydrophobic drugs in the lipid bilayer, and amphiphilic drugs at the interface between the two [79,80,81,82]. Many liposome-derived nanosystems have emerged over the years, such as niosomes and spanlastics. Niosomes range from 10 to 1000 nm and are structurally similar to liposomes since they also consist of an amphiphilic bilayer. However, instead of phospholipids, they are constituted of non-ionic surfactants. This modification has allowed for overcoming the problems that liposomes usually have related to large-scale production, sterilization and physical stability, as niosomes are more stable, economical, biodegradable and easily prepared [83,84,85,86]. In turn, spanlastics are nanovesicular structures with a size of around 180–450 nm, resulting from the modification of niosomes once they also consist of a non-ionic surfactant, Span^®^ 60 (sorbitan monostearate 60). However, what makes them different is the presence of an edge activator, such as polyvinyl alcohol (PVA) or Tween 80, which helps to reduce the size and interfacial tension and confer elasticity to their walls, thus allowing the deformation of the vesicle. This deformation provides these vesicles with the added advantage of increasing their EE and permeation through different physiological membranes [87,88,89,90].

Nanosuspensions are colloidal dispersions in which solid drug nanoparticles are suspended within a liquid with a particle size ranging from 1 to 1000 nm. These nanocarriers have the advantage of being easy to prepare and increasing the permeation and dissolution rates of hydrophobic drugs at the site of action, consequently increasing their therapeutic efficacy [91,92,93,94].

A summary of the characteristics of different nanosystems is shown in Table 1 X.

The mechanisms through which nanosystems deliver drugs to the brain in intranasal administration are yet to be fully elucidated. Nevertheless, some studies suggest that nanosystems with sizes of up to 100 nm are able to be transported through neuronal pathways to the brain (nanosystem + drug will reach it). On the contrary, nanoparticles with sizes above 900 nm cannot and will have to release their cargo since only the drug molecule will be transported through neuronal pathways and reach the brain. Hence, whether the nanosystem itself can reach the brain or not is highly dependent on its size [37,95].

The present review aims to summarize and critically analyze the latest scientific literature regarding the efficacy and safety of nanosystems used in the brain targeting of antipsychotic drugs. Contrary to more general and summarized articles, this paper provides specific data concerning nanosystem composition (drugs and excipients), characterization parameters (such as particle size, polydispersity index (PDI), zeta potential (ZP), EE) and in vivo pharmacokinetics and pharmacodynamics, so that readers will have a clear image on the reported formulations’ true potential when navigating the current scientific literature on the topic. A summary of the included type, composition and characterization parameters of the nanosystems is shown in Table 2.

## 2. 2nd Generation Antipsychotics

### 2.1. Quetiapine Nanoemulsion

Boche et al. [96] evaluated the possibility of improving the brain delivery of quetiapine by developing a nanoemulsion for administration through the IN route. This biphasic dispersion was constituted of the following: an oil, Capmul^®^ MCM (medium-chain mono- and diglycerides); a non-ionic surfactant, Tween^®^ 80, with non-irritant properties for the nasal tissue; a cosurfactant/cosolvent, Transcutol^®^ P, which has the capacity to stabilize the nanoemulsion and increase the solubility of the active ingredient, and in combination with Tween^®^ 80, forms a less viscous nasal preparation; and also a second cosolvent, propylene glycol. The nanoemulsion was characterized for droplet size (144 nm), PDI (0.193), ZP (−8.131 mV) and EE (91%) and exhibited a small and homogeneous droplet size, neutral charge and high drug content, showing adequacy for increased system stability and intranasal administration. They also performed an in vivo pharmacokinetic study in rats, with both intravenous (IV) and IN administration of the developed nanoemulsion and a drug solution. It was observed that the brain concentration of quetiapine after the IN administration of the nanoemulsion was superior to that of the IN solution, resulting in a higher brain bioavailability. Furthermore, the time that it took to reach maximum drug concentration (T_max_) in the brain was shorter for the IN nanoemulsion than for the IV administration, indicating a faster brain transport of quetiapine. Additionally, the plasma area under the curve (AUC) and maximum drug concentration (C_max_) values following IN nanoemulsion administration were smaller and hence advantageous, as they indicate that there was a reduced systemic drug distribution and, consequently, a diminished likelihood of systemic adverse effects. Furthermore, it was determined that the drug reached the brain more effectively and directly with the IN nanoemulsion since the drug targeting efficiency (DTE%) (268%) and the direct transport percentage (DTP%) (64%) values were higher than those of the IN solution (141% and 29%, respectively), confirming the advantage of the IN administration of the developed quetiapine nanoemulsion, allowing an efficient brain targeting of the drug.

### 2.2. Risperidone Spanlastics, Nanoemulsions and Solid Lipid Nanoparticles

Abdelrahman et al. [98] conducted a study to investigate whether a spanlastics nanovesicle was suitable for risperidone brain targeting via intranasal delivery. The nanovesicular structure was made of a non-ionic surfactant (Span^®^ 60), a cosolvent (ethanol) and an edge activator (PVA). The use of PVA was beneficial for reducing the particle size and imparting elasticity to the nanovesicle. Characterization of the prepared nanovesicular system was undertaken, and it showed a mean particle size of 103.4 nm (Figure 3A), a PDI of 0.341 and a ZP of −45.92 mV, all indicative of physical stability and the minimal possibility of aggregation of the prepared spanlastics. The reported EE was 64%, and it showed a relatively high viscosity of 70 cPs due to the presence of two viscous surfactants, with no need for the addition of gelling agents. An in vivo pharmacokinetic study was performed on rats (Figure 3B), and the risperidone concentration in both brain and plasma was determined after the IN administration of the developed spanlastics and then compared to an IN solution of the drug. The spanlastics system exhibited both higher brain C_max_ and T_max_ values than the drug solution, indicating that a greater brain concentration was achieved, but it took longer to reach the brain. In addition, the DTE% (469%) and DTP% (79%) were higher for the IN spanlastics than for the drug solution (217% and 55%, respectively), further indicating the superiority of the developed nanosystem and the high partitioning of the drug to the brain via both the olfactory route and through the BBB (after systemic absorption). Additionally, the developed formulation proved to be safe for intranasal administration, showing high biocompatibility in a histopathological study (sheep nasal mucosa) (Figure 3C).

Furthermore, Đorđević et al. [97] designed two risperidone nanoemulsions for parenteral administration. The biphasic dispersions consisted of medium-chain triglycerides and purified soybean oil (oils), soy lecithin and sodium oleate (emulsifiers), benzyl alcohol (cosolvent), butylhydroxytoluene (antioxidant), glycerol (isotonic agent) and distilled water. Additionally, one of the nanoemulsions also contained polysorbate 80 (hydrophilic surfactant) in its composition. The characterization of the nanoemulsions revealed that both formulations had a favorable droplet size (around 184 nm), uniform size distribution (PDI of approximately 0.11) and high surface charge (ZP around −56 mV), suggestive of nanoemulsion stability with low viscosity (approximately 5 cP), which is especially desirable for IV administration. The results of the pharmacokinetic studies performed on rats after the intraperitoneal administration of either one of the developed nanoemulsions or a drug solution showed that both the brain and plasma AUC values were higher after the administration of the nanoemulsions in comparison to the drug solution. The differences in performance between the two nanoemulsions were not significant, and thus there was no influence due to the incorporation of polysorbate 80 into the composition of the formulation.

On the other hand, Qureshi et al. [99] formulated and optimized chitosan SLN for IN administration. The developed SNL consisted of oleic and stearic acids, distilled water, Tween^®^ 80 (surfactant) and a chitosan solution for coating. A small particle size (132.7 nm), high drug content (8%) and high in vitro drug release (81%) were obtained. In the in vivo pharmacokinetic study in rats, either the developed SLN or a risperidone suspension was administered through the IN or IV routes. The results showed that the SLN produced higher brain C_max_ and AUC values when compared to the IN and IV drug suspensions despite having a higher T_max_ (due to the controlled drug release of the drug from the SLN).

### 2.3. Olanzapine Solid Lipid Nanoparticles, Polymeric Nanoparticles, Nanostructured Lipid Carriers and Niosomes

Joseph et al. [100] conducted a study in which they developed SLNs for the treatment of acute episodes of schizophrenia. Two different types of SLN were formulated, one consisting of glyceryl monostearate (solid lipid with surfactant capability), water and poloxamer 188 (hydrophilic surfactant), and another having the same composition but with an additional coating of Tween^®^ 80, with the intention that this additional coating would provide the SLNs with an increased brain targeting capability, crossing the BBB by means of endocytosis. The characterization of the nanosystems indicated that the presence of Tween^®^ 80 slightly reduced the values of both particle size (from 157.42 to 151.29 nm) (Figure 4A) and PDI (from 0.411 to 0.346). There was also a decrease in the ZP absolute value with the addition of the Tween^®^ 80 coating (from −37.25 to −33.67) owing to the adsorption of the non-ionic surfactant to the SLN surface. Moreover, the high ZP absolute values indicate the potentially good stability of these formulations. The developed nanosystems also had a high EE (73% and 75%) and drug content (around 4%) (without and with Tween^®^ 80, respectively). Given the similarities between the two SLNs, their in vitro drug release profiles were also quite similar, almost overlapping, showing a sustained drug release over a time period of 50 h (Figure 4B). The in vivo pharmacokinetic study performed on rats found an increase in efficacy with the IV administration of both SLNs when compared to a drug solution. In particular, the Tween^®^ 80-coated SLN displayed higher plasma bioavailability when compared to the non-coated SLN and the drug solution (higher AUC and C_max_). In vivo pharmacodynamic studies were also performed on rats. In the apomorphine-induced sniffing and climbing behavior study, the antipsychotic effect of both developed SLNs (inhibition of “continuous sniffing” behavior) was sustained for 48 h, while the antipsychotic effect of the drug solution only lasted 8 h. Additionally, Tween^®^ 80-coated SLN administration showed a lower sniffing score at all time points when compared to the non-coated SLN, evidencing a higher degree of inhibition and, therefore, higher therapeutic efficacy. In the weight gain study (Figure 4C), a formulation was administered to each group of animals daily for 28 days, and the body weight was measured every other day. Animals to whom SLNs were administered demonstrated less weight gain than animals to whom the drug solution was administered, with the Tween^®^ 80-coated SLNs inhibiting weight gain to a greater extent than the non-coated SLNs. Hence, these results indicated that the Tween^®^ 80-coated SLNs successfully delivered olanzapine to the brain, having increased efficacy and potentially enabling the treatment of schizophrenia with reduced drug doses.

Natarajan et al. [101] also developed olanzapine SLNs for IV administration for the purpose of increasing therapeutic efficacy and reducing systemic side effects, allowing for reduced necessary drug doses. These SLNs were composed of tripalmitin (solid lipid), stearyl amine (positive charge inducer), Tween^®^ 80 (surfactant) and water. The developed nanosystem presented with a small and homogeneous particle size (particle size of 110.5 nm, PDI of 0.340), potentially good stability (ZP of +35.29 mV) and high drug content (EE of 96%). The in vivo pharmacokinetic study was performed on rats to whom either the olanzapine SLN or a drug suspension was administered. Following the IV administration of the SLN, the olanzapine plasma concentration was higher than the achieved after the IV administration of the drug suspension. The plasma drug concentration also remained constant until 6 h after SLN administration, whereas with the drug suspension, it reached a minimum value after 3 h due to a prolonged drug release from the SNL. Similarly, the developed SLN led to a 23-fold higher brain bioavailability than the drug suspension after 24 h, and hence the administration of the SLN led to much slower clearance and prolonged drug levels.

On the other hand, Joseph et al. [102] encapsulated olanzapine in NPs, mainly in order to minimize the extrapyramidal adverse effects associated with its administration. Two different NPs were prepared, one with and one without a Tween^®^ 80 coating. The NPs were also made of polycaprolactone and poloxamer 188. The results of the characterization studies demonstrated that both NPs presented with particle sizes of less than 100 nm, thereby allowing a potentially better brain targeting of the nanosystem (73.28 and 81.41 nm, for the non-coated and coated NPs, respectively). The formulations also exhibited a relatively uniform size distribution, with a PDI of 0.231 and 0.312, a high EE of 79 and 77%, and a potentially high physical stability, with a ZP of −32.46 and −27.81 mV (non-coated and coated NP, respectively). The in vivo pharmacokinetic studies involving the IV administration of an olanzapine solution or the developed NPs to rats showed that the NPs led to higher brain and plasma C_max_ and AUC values than the drug solution. Furthermore, the Cmax obtained with the coated NPs was higher than with the uncoated counterpart. Thus, it was concluded that the developed NPs enhanced drug permeation across the BBB and that the coated NPs did so more significantly. The catalepsy pharmacodynamic study was also conducted on rats, with the animals to which the NPs were administered demonstrating a more significant inhibition of catalepsy than the drug solution group and with the coated NPs having a more significant effect.

In turn, Gadhave et al. [103] incorporated olanzapine into NLC for use in intranasal administration. The NLCs were made of Labrafil^®^ M 1944 CS (liquid lipid), Compritol^®^ 888 ATO and Gelucire^®^ 44/14 (solid lipids) and Tween^®^ 80 (surfactant). Gelucire^®^ 44/14 was chosen since it has been shown to be able to inhibit efflux transporters, such as P-glycoprotein, thus potentially increasing brain drug bioavailability. Based on this core composition, they prepared a second formulation by adding two polymers in order to potentially extend the residence time of the preparation in the nasal cavity: hydroxypropyl methylcellulose K4M (HPMC, viscosifying and mucoadhesive agent) and poloxamer 407 (in situ gelling agent), hence making an NLC nanogel. The NLCs had a particle size of 88.95 nm, a PDI of 0.31, a EE of 89%, a drug loading of 6% and a ZP of −22.62 mV, thus having the required characteristics in terms of size, homogeneity, stability and drug content. In the in vivo pharmacokinetic study, the results showed that the IN administration of the NLC nanogel significantly increased the brain concentration of olanzapine when compared with the IV administration of the NLCs. Moreover, IV NLC administration revealed a greater accumulation of the drug at the administration site and in the systemic circulation, thus leading to a higher risk of hematological adverse effects. Therefore, the IN delivery of the NLC nanogel showed a capacity for olanzapine brain targeting. An in vivo safety study was undertaken wherein rats were given low (1 mg/kg), medium (2 mg/kg) and high (4 mg/kg) doses of the IN NLC nanogel over 28 days. The study revealed that all of the hematological parameters were within the normal physiological range, and hence the IN NLC nanogel did not provoke agranulocytosis or leukopenia in the animals, being potentially safe. This could be explained by the fact that the NLC nanogel contains a mucoadhesive polymer in its composition (HPMC) that helps retain the drug in the nasal cavity for a more prolonged time period, reducing the amount of drug that will be absorbed into the bloodstream and reducing the risk of adverse hematological effects. Additionally, a histopathological study of the nasal mucosa conducted under the same circumstances further confirmed that chronic administration of the NLC nanogel is safe since the microscopic images did not reveal any physiological, structural changes compared to the control group. Therefore, the developed NLC nanogel was shown to be both potentially effective and safe.

Khallaf et al. [104] prepared a different type of nanosystem for olanzapine brain targeting: chitosan coated niosomes, to be administered through the IN route. These were composed of a non-ionic surfactant, Span^®^ 80, cholesterol, and a chitosan coating. Characterization of the nanovesicles revealed a particle size of 250.1 nm and an EE of 72%. Subsequently, this group determined, using rats, the olanzapine pharmacokinetic parameters after administration of an olanzapine solution by the IV or IN route, and of the niosomes by the IN route. At first, the IV drug solution exhibited higher brain drug concentrations and AUC values than the IN formulations, which may have been due to the high transport of olanzapine through the BBB, by passive diffusion, as a result of an initially high plasma drug concentration resulting from the IV administration. However, over time the brain drug concentration that resulted from IN noisome administration increased significantly, resulting in significantly higher brain C_max_ and AUC values, when comparing to the IN and IV drug solutions. These findings could be attributed to the controlled release of the drug from the developed nanosystems, which might have led to an increased residence time of the drug in the rat nasal cavity due to the chitosan’s mucoadhesive properties. Since non-ionic surfactants are the predominant components of niosomes, a histopathological study was also carried out to observe whether it caused toxic effects at the application site of the nanovesicles and the nasal mucosa. Nevertheless, there were no signs of irritation, edema, hemorrhage, or necrosis in the analyzed histological structure, and hence the nanosystem was considered safe for IN administration.

### 2.4. Asenapine Nanostructured Lipid Carriers and Nanoemulgel

Singh et al. [105] explored the IN route for brain targeting of asenapine formulated into glycol chitosan coated NLC. The developed nanocarrier was composed of a liquid lipid (oleic acid), a solid lipid (glyceryl monostearate), water, Tween^®^ 80, and glycol chitosan, a product of the conjugation of chitosan with ethylene glycol. This conjugate has the advantage of being soluble at physiological pH, while plain chitosan is only soluble at acidic pH values. The formulation was characterized regarding particle size (184.2 nm), ZP (+18.83 mV), and EE (84%). The in vitro drug release assay showed that the developed NLC had a controlled release profile (Figure 5A), and cell viability studies revealed the biocompatible nature of both drug and excipients (Figure 5B). Plasma and brain pharmacokinetic parameters (Figure 5C), after administration of an asenapine solution, by IV or IN route, or the NLC, via IN route, were determined in rats. No significant differences were observed between the plasma C_max_ and T_max_ values after IN administration of the drug solution and the NLC. However, significant differences were observed concerning IV solution administration, with it leading to a higher plasma C_max_ value, which indicates that IN administration decreased systemic drug exposure. Moreover, it was observed that asenapine brain concentration was higher after NLC IN administration, when compared to the IV and IN solutions, at all time points, being measurable up to 24 h, which is related to a potentially prolonged drug retention at the site of action and, hence, therapeutic effect. The authors justified these results by hypothesizing that the hydrophilic glycol chitosan chain and the existence of a Tween^®^ 80 coating on the NLC surface protected the drug from both enzymatic degradation and uptake by macrophages. Additionally, since asenapine has been reported to have a teratogenic potential in early pregnancy, an embryological and fetal toxicology study was carried out in female rats. In the group exposed to the asenapine solution there was a substantial decrease in the total number of living fetuses, as well as their size, contrarily to the group exposed to the NLC, in which the percentage of fetal malformations was significantly lower. Overall, it was therefore concluded that the encapsulation of asenapine in the developed NLC reduced its teratogenic potential, hence making it a safer option in terms of embryological toxic effects.

On the other hand, Kumbhar et al. [106] performed a study in order to optimize an asenapine mucoadhesive nanoemulsion, to promote its adhesion to the nasal mucosa and improve brain drug targeting. Thus, they developed a nanoemulgel constituted of an oil (Capmul^®^ PG-8), a surfactant (Kolliphor^®^ RH40), a cosurfactant (Transcutol^®^ HP), water and a gelling agent (Carbopol^®^ 971). The nanoformulation exhibited a droplet size of 21.2 nm and a PDI value of 0.355. Although a longer retention time of the formulation in the nasal cavity was intended, the developed nanoemulgel should not affect the normal functioning of the nasal cilia, especially for chronic therapeutic regimen purposes. Hence, the nasal ciliary toxicity of the developed nanoemulgel, as well as its nanoemulsion counterpart (same formulation but without the gelling agent, used for comparison purposes), were evaluated in the nasal mucosa of sheep. Since no morphological changes were observed for either formulation, the authors inferred that the drug and all used excipients were harmless to the nasal mucosa, and hence the developed formulations were safe for IN administration. Next, an in vivo pharmacokinetic study was performed on rats with the IN administration of either the nanoemulgel or the nanoemulsion or the IV administration of a drug solution. The results showed that brain C_max_ was reached faster with IN administration (1 h) when compared to the IV route (3 h). In addition, the IN nanoemulgel showed higher brain C_max_ and AUC values when compared to the IN nanoemulsion and IV solution. The high viscosity and mucoadhesion of the nanoemugel resulted in a prolonged residence time in the nasal cavity and, consequently, in reduced mucociliary clearance, which was reflected in the resulting higher brain drug levels. For the same reason, the IN nanoemulgel evidenced the highest pharmacokinetic ratios values when compared to the IN nanoemulsion, both in what concerns DTP% (80% and 73%, respectively) and DTE% (689% and 517%, respectively). Finally, three pharmacodynamic studies were performed, as follows: the catalepsy test, the induced locomotor activity test, and the paw test. The rats to which the nanoemulgel or the nanoemulsion were administered by the IN route showed a very similar cataleptic response, exhibiting no signs of catalepsy 6 h after drug administration. Locomotor count values were also similar for both formulations, leading to a significant reduction in locomotor activity due to the dopaminergic antagonist effect of asenapine. Particularly, in the paw test, there was no significant alteration in the forelimb retraction time (FRT) of the treated animals, indicating the absence of extrapyramidal adverse effects. Moreover, the prolongation of the hindlimb retraction time (HRT) was also representative of the potential antipsychotic effect of the developed asenapine IN formulations.

### 2.5. Lurasidone Nanostructured Lipid Carriers and Mixed Polymeric Micelles

Jazuli et al. [107] prepared NLCs for IN administration to improve the brain targeting of lurasidone. This developed lipid carrier was composed of a liquid lipid (Capryol^®^ 90, propylene glycol monocaprylate), a solid lipid (Gelot^TM^ 64, a mixture of glycerol monostearate and PEG-75 stearate), a surfactant (Tween^®^ 80) and a cosurfactant (Transcutol^®^ P), all of which were selected based on drug solubility capability and resulting nanosystem stability. The NLCs were characterized according to their particle size (207.4 nm), PDI (0.392) and EE (92%). An in vivo pharmacokinetic study was conducted using rats to determine lurasidone brain distribution after IN NLC administration or after the administration of either an IN drug solution or an oral drug suspension. The developed IN NLCs were found to be more effective in increasing lurasidone brain bioavailability, leading to a higher brain C_max_ and AUC, a lower elimination rate constant and a longer half-life time than the other administered formulations. Moreover, IN administration led to a lower plasma drug concentration, minimizing extravascular drug distribution and, consequently, potential systemic side effects, making it a safer option than other routes of administration.

Pokharkar et al. [108] also developed a lurasidone IN nanosystem, in this case, MPMs. The amphiphilic copolymers of Pluronic^®^ F127 and Gelucire^®^ 44/14 were employed to formulate the MPMs. The developed MPMs exhibited a particle size of 175 nm and an EE of 98%. In the in vivo pharmacokinetic study, either a drug solution or the developed MPMs were administered to rats by both IV and IN routes. The IN administration of the MPMs resulted in high lurasidone brain bioavailability, leading to a higher brain C_max_ and AUC than those obtained with all other formulations and administration routes, thereby indicating the effectiveness of the developed nanosystem in brain drug targeting through the IN route. Moreover, high DTE% (394%) and DTP% (74%) values proved that the IN administration of the developed MPMs allowed lurasidone brain targeting by direct transport through the olfactory and trigeminal nerve pathways.

### 2.6. Zotepine Nanosuspension

Pailla et al. [109] developed a zotepine nanosuspension intending to increase its brain targeting, thereby reducing the needed drug doses and, hence, decreasing dose-dependent side effects. The developed colloidal dispersion consisted of solid drug particles suspended in a solution containing surfactants, namely soy lecithin and Pluronic^®^ F-127, intended to increase drug permeability and decrease particle size and mucus viscosity, and a polymeric and mucoadhesive stabilizer, HPMC E15, with the capacity of increasing formulation stability (suspender function) and increasing the nanosuspension’s residence time in the nasal mucosa. The developed formulation revealed a particle size of 330 nm, a PDI of 0.208 and a ZP of 18.26 mV. The drug concentrations in the rat plasma and brain tissue samples were determined after the administration of either a zotepine solution, IV or IN, or the developed nanosuspension through the IN route. The results showed that the zotepine plasma concentration was highest after IV solution administration, then after IN solution administration, and significantly lower in the case of the IN nanosuspension, proving the IN route to be a potentially safer route with better brain drug targeting. This was confirmed by the brain drug levels since the developed IN nanosuspension displayed higher brain C_max_ and AUC values than the other formulations. Moreover, the IN nanosuspension exhibited the highest DTE% (33,712%) and DTP% (97%) values when compared to the IN solution, which further verified the promising potential of the developed formulation for brain drug targeting when administered through the IN route.

### 2.7. Amisulpride Nanoemulsion and Nanoemulgel

Gadhave et al. [110] produced, optimized, and investigated the therapeutic efficacy of a nanoemulsion and a nanoemulgel for the treatment of schizophrenia. The developed nanoemulsion was composed of an oil (Maisine^®^ CC, glyceryl monolinoleate), a surfactant (Labrasol^®^), a cosurfactant (Transcutol^®^ HP) and water. The nanoemulgel had that same composition except for the addition of the in situ gelling agents of xanthan gum and poloxamer 407, which were added in order to increase the formulation’s retention at the administration site, minimizing the drug loss caused by mucociliary clearance and consequently increasing amisulpride absorption and brain delivery. Both the nanoemulsion and the nanoemulgel were subjected to droplet size (92.15 and 106.11 nm, respectively), PDI (0.46 and 0.51, respectively), ZP (−18.22 and −16.01 mV, respectively) and EE (99%) characterization. To determine the formulations’ pharmacokinetic profile, the nanoemulgel was administered via IN and the nanoemulsion was administered either via IN or IV to the rats. It was observed that the IN administration of both the nanoemulgel and the nanoemulsion led to higher brain drug concentrations than the IV route. Furthermore, the IN nanoemulgel led to the highest brain and lowest plasma C_max_ and AUC values. Thus, the pharmacokinetic data demonstrated that the IN administration of the developed nanoemulgel and nanoemulsion led to direct drug transport to the brain, with the nanoemulgel being the most effective. Additionally, the therapeutic efficacy of the developed amisulpride nanoformulations was also determined by the catalepsy test, the induced locomotor activity test and paw test studies. It was found that the rats treated with either the nanoemulgel or the nanoemulsion for 28 days did not demonstrate symptoms of catalepsy (compared to the control group). However, the nanoemulgel led to a lower catalepsy response than the nanoemulsion. This result could be justified by the rapid permeation and drug-modified release from the developed formulation. Furthermore, both formulations led to a significant reduction in locomotor activity, proving that the drug did, in fact, reach the intended therapeutic target, having antagonistic action on the brain’s dopamine D2 receptors. Additionally, the nanoemulgel showed a greater reduction in locomotor activity, therefore exhibiting greater therapeutic efficacy. Furthermore, the paw test revealed no major changes in the FRT; however, there was an extensive reduction in HRT in the animals to which the IN nanoemulgel or the IN nanoemulsion were administered, which reinforces the absence of extrapyramidal effects and the existence of therapeutic efficacy. Finally, the occurrence of toxicological symptoms and signs was evaluated through hematological and pathology analyses. In these studies, the authors concluded that the IN nanoemulgel did not cause any hematological toxicity, whereas the IV nanoemulsion led to dose-dependent toxic effects, leading to the deaths of those animals in which doses of 5 mg/kg were administered, which was further proof of the higher safety of the IN route as well as the superiority of the developed amisulpride nanoemulgel.

### 2.8. Clozapine Nanosuspension

In light of all the problems associated with clozapine oral administration, such as low solubility and low dissolution rates, high gastrointestinal and hepatic metabolism and consequent low cerebral bioavailability, Patel et al. [111] developed a clozapine nanosuspension for IN administration. The nanosuspension was prepared by dispersing solid powder particles of the drug in deionized water containing D-α-tocopheryl polyethylene glycol succinate 1000 (TPGS) and polyvinylpyrrolidone K30 as surfactants and formulation stabilizers. TPGS also has the reported ability to inhibit efflux transporters, such as P-glycoprotein, thus increasing drug bioavailability. The nanosuspension exhibited a small particle size (281 nm), thus leading to a high dissolution rate and surface area for permeation, and a ZP of −0.83 mV, due to the non-ionic nature of TPGS, adsorbed to the surface of the particles. The in vitro drug release assay showed that the nanosuspension had a faster and overall higher drug release than that of a conventional drug suspension (Figure 6A). The in vivo pharmacokinetic study using rats (Figure 6B) demonstrated that the IN nanosuspension substantially increased clozapine brain concentration when compared to the conventional oral suspension, displaying significantly higher C_max_ and AUC values. Additionally, a nasal ciliary toxicity study proved the suitability of the nanosystem for IN administration (Figure 6C).

## 3. 3rd Generation Antipsychotics

### Aripiprazole Nanoemulgel

Kumbhar et al. [112] formulated an aripiprazole nanoemulgel for direct drug transport to the brain via IN delivery. To do this, a nanoemulsion was first developed containing Capmul^®^ PG-8 as the oil, TPGS and Transcutol^®^ HP as the surfactant and cosurfactant, respectively, and distilled water. The nanoemulsion was then converted into a nanoemulgel by adding Carbopol^®^ 971 as a gelling agent to the external phase in order to increase the nanoemulsion’s viscosity and, hence, formulation suitability for IN administration. The developed nanoemulgel exhibited a droplet size of 121.8 nm, a PDI of 0.248 and a ZP of −18.89 mV. Pharmacokinetic studies were conducted on rats with the nanoemulgel being administered via the IN route, while the aripiprazole nanoemulsion was administered via the IV and IN routes. The superiority of the IN route was evident due to the IN nanoemulgel and IN nanoemulsion leading to a higher brain C_max_ than the IV nanoemulsion. Moreover, the IN nanoemulgel led to a higher brain C_max_ than the IN nanoemulsion. The DTP% (90% and 77%, respectively) and DTE% (97% and 83%, respectively) were also found to be higher for the IN nanoemulgel, when compared to the IN nanoemulsion, evidencing the greater capability of the nanoemulgel for brain drug targeting. Additionally, three pharmacodynamic studies were also performed: catalepsy, induced locomotor activity and paw tests, in which either the nanoemulgel or the nanoemulsion were administered intranasally. No signs of catalepsy were observed within 3 h after administration of either formulation, but by the 6th hour, a catalepsy effect was observed for the nanoemulsion, which might be explained by the fact that this formulation was eliminated by mucociliary clearance, while the nanoemulgel remained in the nasal cavity for a longer time period due to its mucoadhesive properties. Furthermore, both formulations were equally effective in locomotor activity reduction, demonstrating that aripiprazole was effective in both formulations in partially blocking dopaminergic receptors, thus resulting in locomotor activity decrease. Furthermore, it was found that there was no significant change in FRT in the animals treated with either IN nanosystem, indicating that there were no extrapyramidal symptoms resulting from their administration. Moreover, for the groups in which the IN nanoemulgel was administered, there was a significant increase in HRT when compared to the IN nanoemulsion. Additionally, no ciliary toxicity was observed in sheep nasal mucosa tissue following nanoemulgel or nanoemulsion exposure, indicating the safety of the developed formulations and their components for IN administration.

## 4. Final Discussion

It is noteworthy that although all of the included studies aimed to develop an innovative nanosystem for the brain targeting of antipsychotic drugs, great heterogeneity exists in what concerns the selected molecule itself and the developed nanosystem type (and composition). Thus, it becomes difficult to compare them directly without the risk of introducing great bias into the data interpretation. Even in the case of the most extensively studied antipsychotic drug, olanzapine [100,101,102,103,104], the developed nanosystem type differed, including SLN, NP, NLC and niosomes. Nevertheless, some general conclusions can be drawn regarding a few of the evaluated parameters.

Particle/droplet size has a significant impact on drug delivery through the IN route. Nanosystems have to be smaller than 200 nm in size for direct transport to the brain via the neuronal pathways [113]. Most of the nanosystems developed in the included studies exhibited a size smaller than that mentioned above, except for the nanosuspensions developed by Pailla et al. (330 and 519.26 nm) [109], the NP developed by Qureshi et al. (281 nm) [99] and the niosomes developed by Khallaf et al. (250.1 nm) [104]. The asenapine nanoemulgel developed by Kumbhar et al. [106] had the smallest droplet size of only 21.2 nm. Additionally, particle size distribution is characterized by the PDI value, measured on a scale of 0 to 1. PDI values lower than 0.3 represent a monodisperse particle population with high homogeneity [106,111]. The risperidone nanoemulsions developed by Đorđević et al. [97] revealed the lowest PDI value (0.11), suggesting that it is the particle population with the most homogeneous size distribution. Furthermore, nanosystem characterization is not complete without the ZP value, which relates directly to the stability of the nanosystems, indicating the degree of repulsion between adjacent systems. A nanosystem with a high absolute value of ZP will have a lower propensity for forming aggregates, thus exhibiting greater stability. According to the generality of scientific literature, a nanosystem is stable if it has a value either greater than +30 mV or lower than −30 mV. Additionally, positively charged nanosystems have the advantage of interacting with the negatively charged sialic acid groups present in the nasal mucosa, thereby increasing the retention time and decreasing drug clearance [101,105]. Out of the seventeen studied nanosystems, only four exhibited a positive ZP value, with the olanzapine SLNs [101] having the highest ZP value (+35.29 mV) due to the presence of stearyl amine (positive charge inducer) in the formulation’s composition. Out of the nanosystems that had a negative ZP, the risperidone nanoemulsions [97] had the lowest value (−56 mV) due to the presence of sodium oleate (anionic surfactant) and, consequently, greater potential for stability.

The antipsychotics’ EE was most often determined by the indirect method, using the following formula:$$EE\;\left(\%\right)=\frac{W1-W2}{W2}\times 100$$
where *W*1 is the total amount of antipsychotic drug used in the formulation preparation, and *W*2 is the amount of free antipsychotic drug remaining in the solution. Comparing the obtained *EE* values, the amisulpride nanoemulgel [110] displayed a higher *EE* than the other formulations (99%), providing the possibility for a greater amount of drug to be encapsulated and, consequently, available to reach the intended therapeutic site.

To quantify the extent of the drug reaching the brain after the *IN* administration of the formulated antipsychotic drugs, the authors used two pharmacokinetic ratios, *DTE*% and *DTP*%. The *DTE*% was calculated according to the following formula:$$DTE\%=\frac{\left(\frac{AUC\;brain}{AUC\;plasma}\right)IN}{\left(\frac{AUC\;brain}{AUC\;plasma}\right)IV}\times 100$$

The *DTE*% value can range from 0 to +∞ and quantifies a drug’s overall tendency to accumulate in the brain after *IN* versus *IV* administration. Values greater than 100% reveal that there is a more effective brain targeting of the drug via the *IN* route than via the *IV* route (or another parenteral route) [43].

On the other hand, *DTP*% determines the relative amount of drug reaching the brain via direct pathways, the neuronal olfactory and trigeminal transport pathways, and is calculated as follows:$$DTP\%=\frac{B_{IN}-B_{X}}{B_{IN}}\times 100\;\;\;\;where\;\;\;B_{X}=\frac{B_{IV}}{P_{IV}}\times P_{IN}$$
where *B_X_* is the fraction of brain *AUC* that comes from indirect transport (after *IN* administration, the drug entered the systemic circulation and subsequently crossed the BBB); B_IN_ is the overall brain *AUC* after *IN* administration; B_IV_ is the brain *AUC* after *IV* administration; *P_IV_* is the plasma *AUC* after *IV* administration; and P_IN_ is the plasma *AUC* after *IN* administration. The *DTP*% can theoretically range from −∞ to 100%, with negative *DTP*% values indicating that brain targeting of the drug was most effective when administered via the parenteral route, and hence direct brain drug transport was not significant [43].

Overall, the *DTE*% and *DTP*% values of the studied *IN* nanosystems were higher than those of the solutions/dispersions administered by the same route. In addition, most of the articles that determined the *DTE*% value presented values higher than 100%, proving that *IN* administration is more effective in brain targeting than *IV*. Additionally, all of the nanosystems showed positive *DTP*% values, meaning that these systems can deliver antipsychotic drugs to the brain via direct pathways, bypassing the BBB. Among the studied articles, the zotepine nanosuspension [109] was the nanosystem with the highest values of these ratios (*DTE* 33,712% and *DTP* 97%).

## 5. Conclusions and Future Perspectives

Over the last decade, the incidence of psychotic disorders has increased and consequently, so have antipsychotic drug prescriptions. Such disorders are characterized by a deterioration in the patient’s emotional, cognitive and social functions, which may lead to long-term disability. However, BBB impermeability, P-glycoprotein efflux mechanisms, the low aqueous solubility of the drugs, extensive first-pass metabolism and the broad extravascular distribution of antipsychotics remain significant contributing factors to limited efficacy and numerous adverse effects. Thus, there is a pressing need to develop new strategies to improve the transport of antipsychotics toward the brain. Nanosystems demonstrated through in vivo studies to be a valuable asset in the brain targeting of antipsychotic drugs, especially when administered via the IN route, as they bypass the BBB, reaching the brain directly through the neuronal pathways, thereby increasing the antipsychotics’ therapeutic efficacy. Such nanosystems are expected to reduce the administered dose, the number/frequency of administrations and the occurrence/intensity of adverse effects (sedation, weight gain, extrapyramidal and hematological adverse effects, etc.). Nevertheless, further studies are required, as the exact mechanisms involved in the passage of the nanosystems from the nasal mucosa to the olfactory and trigeminal nerves and subsequently to the CNS, as well as their distribution and interaction with receptors in different parts of the brain, are not fully known and understood. Moreover, further examination of the toxicological effects of nanosystems via IN administration, not only restricted to the nasal mucosal but also on neuronal pathways and inclusively the brain, will also be important before transposition to clinical trials. In sum, the IN administration of nanosystems has the potential to improve the quality of life of patients and help better understand the pathophysiology behind psychotic disorders; however, there is more work to be undertaken before these technologies can proceed to the drug market.

## Figures and Tables

**Figure 1 pharmaceutics-15-00678-f001:**
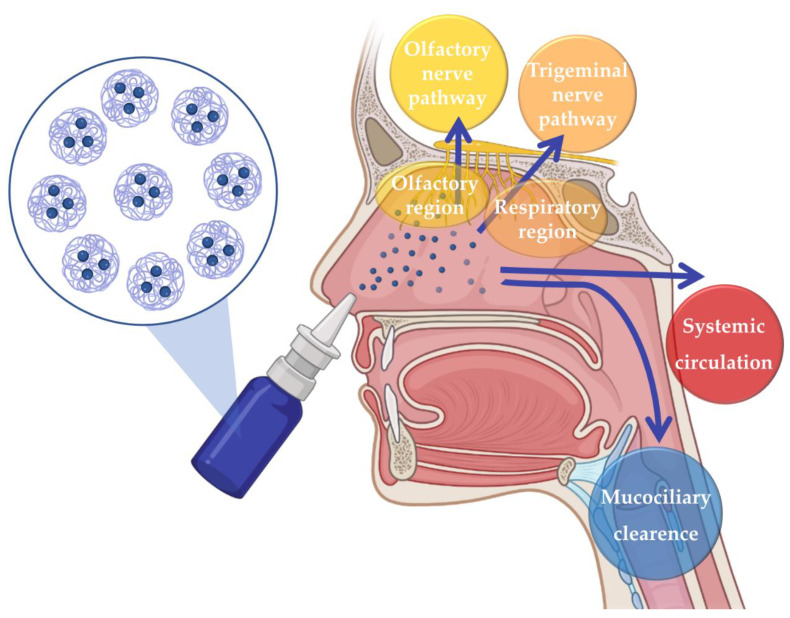
Neuronal and systemic pathways of brain transportation of drugs following intranasal administration.

**Figure 2 pharmaceutics-15-00678-f002:**
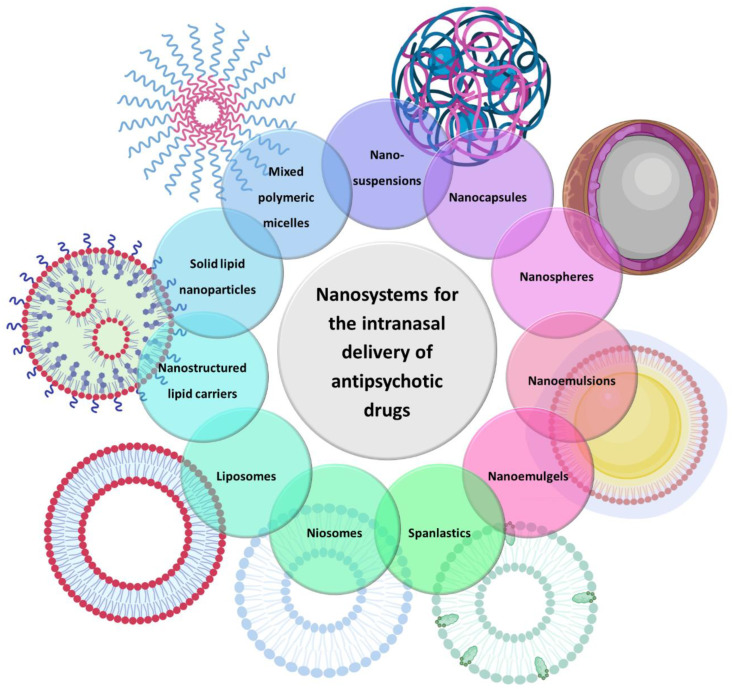
Types of nanosystems that have been used for the brain delivery of antipsychotic drugs.

**Figure 3 pharmaceutics-15-00678-f003:**
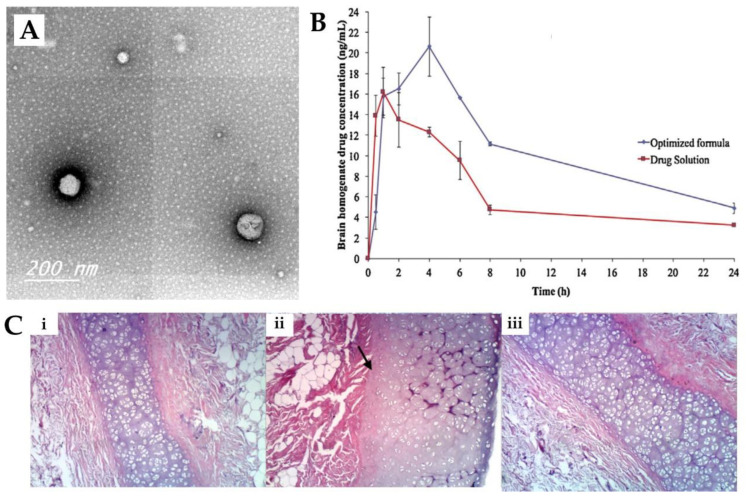
(**A**) Transmission electron microscopy micrographs of the developed risperidone spanlastics; (**B**) Brain concentration vs time profiles of the intranasal risperidone spanlastics (“optimized formula”) and intranasal risperidone solution (“drug solution”); (**C**) Nasal sheep mucosa histopathological evaluation of the developed risperidone spanlastics (**iii**), compared to normal saline (negative control, (**i**)) and isopropyl alcohol (positive control, (**ii**)); adapted from Abdelrahman et al. [98], reproduced with permission from Elsevier [License Number 5477081208011].

**Figure 4 pharmaceutics-15-00678-f004:**
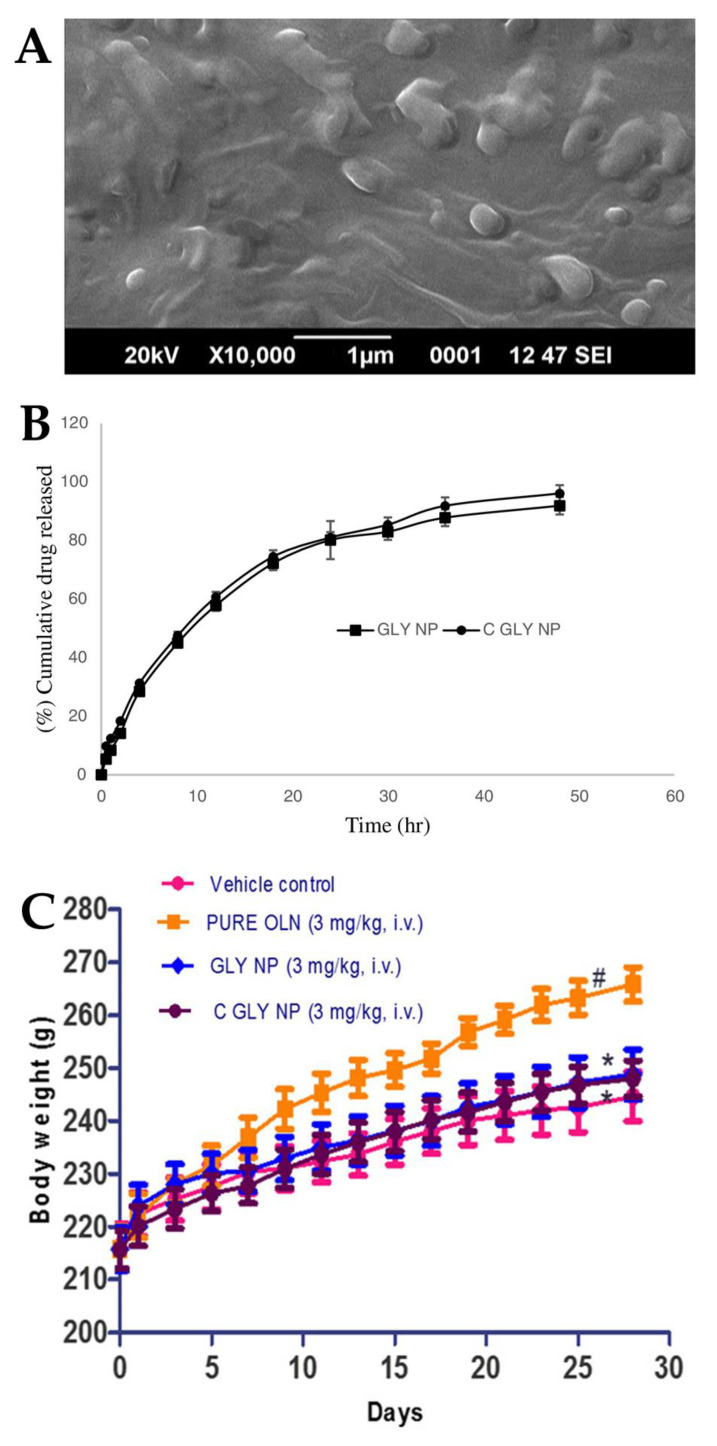
(**A**) Scanning electron microscope image of the developed olanzapine SLN; (**B**) In vitro drug release profiles of the optimized formulations, Tween^®^ 80 coated SLN (“GLY NP”) and non-coated SLN (“C GLY NP”); (**C**) Effect of IV formulation administration, namely olanzapine Tween^®^ 80 coated SLN (“C GLY NP”), olanzapine non-coated SLN (“GLY NP”), olanzapine solution (“PURE OLN”), and drugless vehicle (“Vehicle control”), on animal body weight; adapted from Joseph et al. [100], reproduced with permission from Elsevier [License Number 5477090054162].

**Figure 5 pharmaceutics-15-00678-f005:**
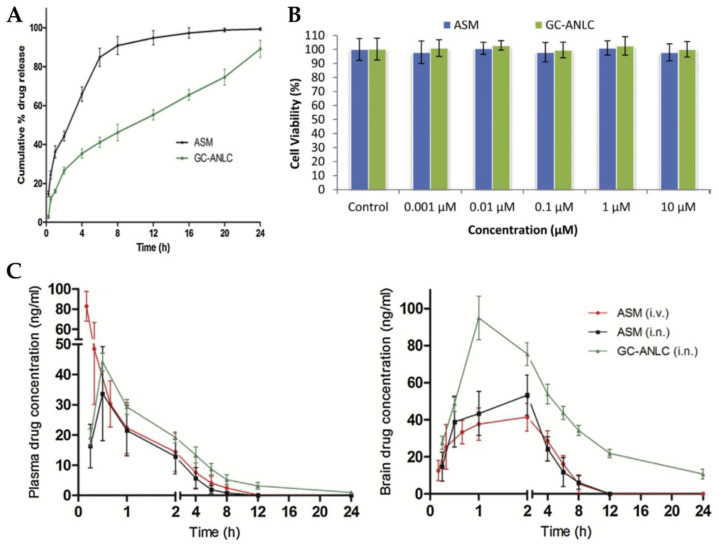
(**A**) In vitro drug release profile of the developed asenapine NLC (“GC-ANLC”) and a drug solution (“ASM”); (**B**) Cell viability results (A549 cells) of the developed asenapine NLC (“GC-ANLC”) and a drug solution (“ASM”); (**C**) Plasma (left) and brain (right) drug concentration vs time profiles of the developed asenapine NLC after IN administration [“GC-ANLC (i.n.)”], and after IV [“ASM(i.v.)] or IN [“ASM(i.n.)] administration of a drug solution; adapted from Singh et al. [105], reproduced with permission from Elsevier [License Number 5477090411361].

**Figure 6 pharmaceutics-15-00678-f006:**
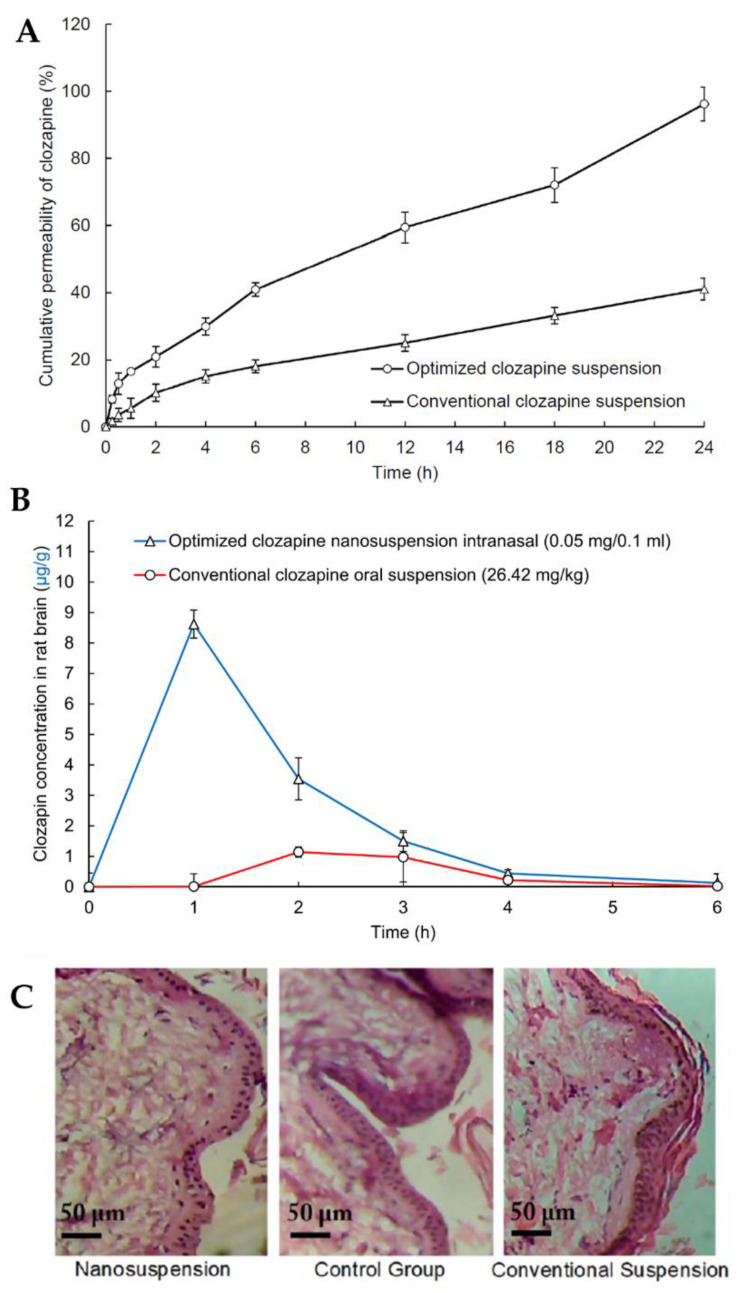
(**A**) Clozapine in vitro drug release profiles of the optimized clozapine nanosuspension and a conventional clozapine suspension; (**B**) Pharmacokinetic clozapine brain profiles after IN administration of the optimized nanosuspension and oral administration of a conventional suspension; (**C**) Histopathology images of nasal tissues after exposure to the optimized clozapine nanosuspension (left) and conventional clozapine suspension (right), compared to a control group (middle); adapted from Patel et al. [111], reproduced with permission from Elsevier [License Number 5477090677061].

**Table 1 pharmaceutics-15-00678-t001:** Main types of nanosystems and respective size range and main compositions, characteristics and advantages.

Nanosystem Type	Size Range	Main Composition	Main Characteristics	Main Advantages	References
Mixed polymeric micelles	20 to 200 nm	Amphiphilic copolymers	Hydrophobic inner core and a hydrophilic outer layer	High stability and encapsulation efficiency, good for the encapsulation of hydrophobic drugs	[53,54,55,56,57]
Nanocapsules	10 to 1000 nm	Liquid lipids + polymers	Reservoir systems with liquid lipid core surrounded by a thin polymeric membrane	Controlled drug release, targeted drug delivery and enhanced drug permeation	[58,59,60,61,62]
Nanospheres		Polymers	No differentiated core, being formed by a dense polymeric matrix		
Solid lipid nanoparticles	50 to 1000 nm	Solid lipids + surfactants	Solid lipid core stabilized by a surfactant layer	Good for encapsulation of hydrophobic drugs, biocompatibility, controlled drug release	[63,64,65,66]
Nanostructured lipid carriers		Solid lipids + liquid lipds + surfactants	Liquid lipids included in an unstructured solid lipid matrix	Higher stability and encapsulation efficiency and lower tendency for crystallization than solid lipid nanoparticles, good for encapsulation of hydrophobic drugs, biocompatibility, controlled drug release	[67,68,69,70]
Nanoemulsions	10 to 200 nm	Oils, surfactants, cosurfactants, water	Liquid-in-liquid colloidal dispersions	Ease of preparation, enhanced drug permeation and increased drug solubility	[71,72,73,74]
Nanoemulgels		Oils, surfactants, cosurfactants, water, gelling polymer	Nanoemulsions with gel characteristics	High viscosity, ideal for retention at the administration site	[75,76,77,78]
Liposomes	20 to 1000 nm	Amphiphilic phospholipids	Spherical colloidal vesicles composed of one or more amphiphilic phospholipid bilayers delimiting an aqueous inner core	Versatility, since hydrophobic drugs can be incorporated into the membrane and hydrophilic drugs can be solubilized within the core, biodegradable	[79,80,81,82]
Niosomes	10 to 1000 nm	Amphiphilic phospholipids + non-ionic amphiphilic surfactants	Modification of liposomes	Higher stability than liposomes, biodegradable	[83,84,85,86]
Spanlastics	180 to 450 nm	Amphiphilic phospholipids + non-ionic amphiphilic surfactant Span^®^ 60 (sorbitan monostearate 60) + edge activator	Modification of niosomes	Small size, deformable, increased encapsulation efficiency and permeation and biodegradable	[87,88,89,90]
Nanosuspensions	1 to 1000 nm	Liquid vehicle + solid particles	Colloidal dispersions in which solid drug nanoparticles are suspended within a liquid	Ease of preparation, increased permeation/dissolution rate of hydrophobic drugs	[91,92,93,94]

**Table 2 pharmaceutics-15-00678-t002:** Summary of the included nanosystems’ type, composition, and characterization parameters.

Drug	Nanosystem Type	Administration Route	Excipients	Droplet Size (nm)	PDI	ZP (mV)	EE (%)	References
Quetiapine	Nanoemulsion	IN	Capmul^®^ MCM, Tween^®^ 80, Transcutol^®^ P, water	144	0.193	−8.1	91	[96]
Risperidone	Nanoemulsion	IV	Medium-chain triglycerides, soybean oil, soy lecithin, sodium oleate, benzyl alcohol, butylhydroxytoluene, glycerol, polysorbate 80, water	184	0.110	−56.0	NR	[97]
	Spanlastics	IN	Span^®^ 60, ethanol, PVA	103	0.341	−45.9	64	[98]
	SLN	IN	Oleic acid, stearic acid, Tween^®^ 80, chitosan, water	133	0.200	+11.8	70	[99]
Olanzapine	SLN	IV	Water, glyceryl monostearate, poloxamer 188,	157	0.411	−37.3	73	[100]
			Water, glyceryl monostearate, poloxamer 188, Tween^®^ 80	151	0.346	−33.7	75	
	SLN	IV	Tripalmitin, stearyl amine, Tween^®^ 80, water	111	0.340	+35.3	96	[101]
	NP	IV	Polycaprolactone, poloxamer 188	73	0.231	−32.5	79	[102]
			Polycaprolactone, poloxamer 188, Tween^®^ 80	81	0.312	−27.8	77	
	NLC	IN	Labrafil^®^ M 1944 CS, Compritol^®^ 888 ATO, Gelucire^®^ 44/14, Tween^®^ 80, HPMC K4M, poloxamer 407	89	0.310	−22.6	89	[103]
	Niosomes	IN	Span^®^ 80, cholesterol, chitosan	250	NR	NR	72	[104]
Asenapine	NLC	IN	Oleic acid, glyceryl monostearate, water, Tween^®^ 80, glycol chitosan	184	NR	+18.8	84	[105]
	Nanoemulgel	IN	Capmul^®^ PG-8, Kolliphor^®^ RH40, Transcutol^®^ HP, water, Carbopol^®^ 971	21	0.355	−14.1	NR	[106]
Lurasidone	NLC	IN	Capryol^®^ 90, Gelot^TM^ 64, Tween^®^ 80, Transcutol^®^ P	207	0.392	NR	92	[107]
	MPM	IN	Pluronic^®^ F127, Gelucire^®^ 44/14	175	NR	NR	98	[108]
Zotepine	Nanosuspension	IN	Soy lecithin, Pluronic^®^ F-127, HPMC E15	330	0.208	+18.3	NR	[109]
Amisulpride	Nanoemulsion	IN	Maisine^®^ CC, Labrasol^®^, Transcutol^®^ HP, water	92	0.460	−18.2	99	[110]
	Nanoemulgel		Maisine^®^ CC, Labrasol^®^, Transcutol^®^ HP, water, xanthan gum, poloxamer 407	106	0.510	−16.0	99	
Clozapine	Nanosuspension	IN	TGPS, polyvinylpyrrolidone K30, water	281	NR	−0.8	NR	[111]
Aripiprazole	Nanoemulgel	IN	Capmul^®^ PG-8, TPGS, Transcutol^®^ HP, water, Carbopol^®^ 971	122	0.248	−18.9	NR	[112]

EE: encapsulation efficiency; HPMC: hydroxypropyl methylcellulose; IN: intranasal; IV: intravenous; MPM: mixed polymeric micelles; NLC: nanostructured lipid carriers; NP: nanoparticles; NR: not reported; PDI: polydispersity index; PVA: polyvinyl alcohol; SLN: solid lipid nanoparticles; TGPS: D-α-tocopheryl polyethylene glycol succinate 1000; ZP: zeta potential.

## Data Availability

Not applicable.

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
