# Peer review of "Nanosystems for Brain Targeting of Antipsychotic Drugs: An Update on the Most Promising Nanocarriers for Increased Bioavailability and Therapeutic Efficacy"

_pharmaceutics, 2023, doi:10.3390/pharmaceutics15020678_

Round 1

Reviewer 1 Report

This is a very nice, easy-to-read and comprehensive review on nanocarriers applied to deliver antipsycotic drug.

I have only 2 comments:

1) write a small paragraph of introduction for point 2 and 3 describing what do you mean as antipsycotic of second and third generation.

2) Include a table drug-driven rporting the differnt carriers used, EE, size, .... 

Author Response

We thank the reviewer for their kind comments. All suggestions and corrections have been taken into account (changes marked in blue in the revised version of the manuscript), and a point-by-point answer in given below.

1) write a small paragraph of introduction for point 2 and 3 describing what do you mean as antipsycotic of second and third generation.

R: We thank the reviewer for their comment, a paragraph with this information has been added from lines 63 to 77.

2) Include a table drug-driven rporting the differnt carriers used, EE, size, ....

R: Thank you for the suggestion, a Table containing this information has been added (Table 2).

Reviewer 2 Report

pharmaceutics-2215353

Nanosystems for brain targeting of antipsychotic drugs: an update on the most promising nanocarriers for increased bioavailability and therapeutic efficacy

The manuscript by Ferreira et al. summarized recent advances in nanosystems for brain targeting of antipsychotic drugs via nose-to-brain transport. There are several recommendations to improve the manuscript, as follows.

1. Similar reviews are available in the literature. Therefore, in the Introduction part, the authors should clarify the novelty and contribution of this review, and explain the differences between this review and previous ones.

2. The authors should include a paragraph clarifying the benefits of nanosystems over conventional formulations (such as gels, solutions, and suspensions) in nose-to-brain delivery.

3. Section 1.1- Nanosystems: The authors should include a table to summarize the primary features, advantages, and disadvantages of each nanosystem in nose-to-brain delivery.

4. Section 1: The authors should expand the part of antipsychotic drug classification. Please clarify the differences, characteristics, and relevant information of the three generations.

5. Figure 2: Structures of some nanosystems are presented. The authors should modify this figure by clarifying the different parts of each structure presented. Please avoid overlapping among images (such as SLNs, NLCs, and the structure on the left).

6. Lines 727 – 279: “Among the studied articles, the amisulpride nanoemulgel [107] was the nanosystem with the highest values of these ratios (DTE 96.98% and DTP 89.73%)”. The statement is incorrect. The authors should re-check the data. It should be noted that the DTE% of 96.98% (<100%) means the system was not effective for brain targeting. Also, DTE% = 96.98% means DTP% = -3.1%. The authors should re-check other articles included in this review. It is noted that many studies reported incorrect DTE% and DTP% values (see review #40 in the reference list, which pointed out many cases with incorrect DTE% and DTP% and how to correct them).

7. In section 1, there is only one subsection (1.1), and there is no 1.2.

Author Response

We thank the reviewer for their comments. All suggestions and corrections have been taken into account (changes marked in blue in the revised version of the manuscript), and a point-by-point answer in given below.

  1. Similar reviews are available in the literature. Therefore, in the Introduction part, the authors should clarify the novelty and contribution of this review, and explain the differences between this review and previous ones.

R: We thank the reviewer for their comment. This clarification has now been added on lines 246 to 252.

  1. The authors should include a paragraph clarifying the benefits of nanosystems over conventional formulations (such as gels, solutions, and suspensions) in nose-to-brain delivery.

R: We thank the reviewer for their comment, a paragraph on the matter has now been added from lines 146 to 151.

  1. Section 1.1- Nanosystems: The authors should include a table to summarize the primary features, advantages, and disadvantages of each nanosystem in nose-to-brain delivery.

R: Thank you for the suggestion, a Table containing this information has been added (Table 1).

  1. Section 1: The authors should expand the part of antipsychotic drug classification. Please clarify the differences, characteristics, and relevant information of the three generations.

R: We thank the reviewer for their comment, a paragraph with this information has been added from lines 63 to 77.

  1. Figure 2: Structures of some nanosystems are presented. The authors should modify this figure by clarifying the different parts of each structure presented. Please avoid overlapping among images (such as SLNs, NLCs, and the structure on the left).

R: Thank you for your comments. In this case, the image for SLN and NLC is the same due to their similarity. Figure was meant to give readers a general image of the different types of nanosystems, without specifying each type of nanosystems’ composition or characteristics. Nevertheless, given your suggestion we have now added a Table summarizing the main types of nanosystems and respective size range and main composition, characteristics and advantages. This will give readers a better and fast understanding of the main differences between the different nanosystem types, without them having to read the text (Table 1).

  1. Lines 727 – 279: “Among the studied articles, the amisulpride nanoemulgel [107] was the nanosystem with the highest values of these ratios (DTE 96.98% and DTP 89.73%)”. The statement is incorrect. The authors should re-check the data. It should be noted that the DTE% of 96.98% (<100%) means the system was not effective for brain targeting. Also, DTE% = 96.98% means DTP% = -3.1%. The authors should re-check other articles included in this review. It is noted that many studies reported incorrect DTE% and DTP% values (see review #40 in the reference list, which pointed out many cases with incorrect DTE% and DTP% and how to correct them).

R: We thank the reviewer for their comment and for noticing the error, this has now been corrected.

  1. In section 1, there is only one subsection (1.1), and there is no 1.2.

R: We thank the author for noticing this, and have divided the introduction section into two subsections, 1.1 and 1.2.

Reviewer 3 Report

Dear Authors,

Thank you for your submission to MDPI Pharmaceutics. Your review is clear, comprehensive, and includes useful graphics for guiding the reader's understanding.

A few very small considerations:

Line 225-226: "t was observed that the brain concentration of quetiapine after IN administration of the nanoemulsion was supe-rior to that of the IN solution, resulting in a higher brain bioavailability" Is the second "IN" intended to read "IV"? In line 223, IV administration is mentioned.
Similarly, in line 235, "values were higher than those of the IN solution" seems to be a typo where "IN" should be "IV".

In general, percentages do not need to be reported with two places after the decimal as I doubt the second place after the decimal would be considered a significant figure.

The Review reads a bit too much like a laundry list of reports in the field. While this is acceptable, the addition of a few "overarching sections", placed either at the beginning or at the end, may improve the overall quality of the review. Such additions may include sections such as:
-- Status of clinical trials
-- Cell- or tissue-specific targeting approaches
-- Transport into the brain, or what happens to the particles once they enter the neural parenchyma
-- Promising revolutionary ideas in the field?

Again, the above topics sections are not required, but may improve the overall use of the review.

Also it would be potentially useful if authors included a chart or table of various nanosystems reviewed, as well as a description of what they are physically (lipid bi-layer with etc. etc.; solid polymer NP with etc. etc.; ...) their advantages, their disadvantages, a list drugs which have been incorporated into that nanosystem, some useful references for interested readers, etc. Such a table would serve as an easy-to-access reference for interested readers.

In general, an excellent submission! I definitely consider the review a useful resource for the research community.

Author Response

We thank the reviewer for their insightful and positive comments. All suggestions and corrections have been taken into account (changes marked in blue in the revised version of the manuscript), and a point-by-point answer in given below.

Line 225-226: "t was observed that the brain concentration of quetiapine after IN administration of the nanoemulsion was supe-rior to that of the IN solution, resulting in a higher brain bioavailability" Is the second "IN" intended to read "IV"? In line 223, IV administration is mentioned. Similarly, in line 235, "values were higher than those of the IN solution" seems to be a typo where "IN" should be "IV".

R: We thank the reviewer for their question. This is indeed correct, since the authors of the mentioned study compared the IN administration of the developed nanosystem to both and IV solution, and an IN solution. Hence, both were done, in order to not only compare the efficacy of the IN route to the IV route (comparison of different administration routes), but also to compare the efficacy of drug incorporation into the nanosystem (hence same administration route, in this case IN). Additionally, DTE and DTP can only be calculated for non-systemic routes, because the AUC obtained from IV administration will already enter the calculation (as can be seen by the presented formula, on page 22.

In general, percentages do not need to be reported with two places after the decimal as I doubt the second place after the decimal would be considered a significant figure.

R: We thank the reviewer for the suggestion and agree, and hence have altered all percentages by presenting the values with no decimal places.

The Review reads a bit too much like a laundry list of reports in the field. While this is acceptable, the addition of a few "overarching sections", placed either at the beginning or at the end, may improve the overall quality of the review. Such additions may include sections such as:

-- Status of clinical trials

-- Cell- or tissue-specific targeting approaches

-- Transport into the brain, or what happens to the particles once they enter the neural parenchyma

-- Promising revolutionary ideas in the field?

Again, the above topics sections are not required, but may improve the overall use of the review.

R: We thank the reviewer for their suggestions. In what concerns drug transport to the brain, we have completed the information by adding a paragraph from line 236 to 243. Even though the suggested topics are not within the scope of our review, we have searched for clinical data on the administration of antipsychotic drugs within nanosystems. Nevertheless, no studies have been done yet, and hence only preclinical data will have to serve as basis for current knowledge. Nevertheless, we consider this to be quite relevant, and hope that science progresses in this way, and that soon these nanosystems will have their true efficacy evaluated in humans.

Also it would be potentially useful if authors included a chart or table of various nanosystems reviewed, as well as a description of what they are physically (lipid bi-layer with etc. etc.; solid polymer NP with etc. etc.; ...) their advantages, their disadvantages, a list drugs which have been incorporated into that nanosystem, some useful references for interested readers, etc. Such a table would serve as an easy-to-access reference for interested readers.

R: Thank you for the suggestions, two Tables have been added containing this information (Tables 1 and 2).

Reviewer 4 Report

This manuscript reviews the current state of antipsychotic drug delivery for the treatment of psychotic disorders such as schizophrenia and bipolar disorder. The authors highlight the issue of adverse drug reactions leading to patient non-compliance with oral administration, and the potential benefits of intranasal delivery. They also discuss the recent advancements in the development of nanosystems for delivering antipsychotic drugs, including polymeric nanoparticles, polymeric mixed micelles, and others. The authors conclude that the intranasal delivery of nanosystems shows promise in improving brain bioavailability and therapeutic outcomes in animal models. Overall, this manuscript provides a comprehensive overview of the current state of antipsychotic drug delivery and the potential for future therapies.

However, several improvements can be made to the manuscript. Firstly, the authors need to define what is meant by "1st, 2nd, and 3rd generations of antipsychotics." This will help readers understand the context of the study better. Secondly, the mechanism of how nanosystems deliver drugs into the brain through intranasal administration should be more thoroughly explained. This will provide readers with a deeper understanding of the science behind the proposed method of delivery.

Author Response

We thank the reviewer for their comments. All suggestions and corrections have been taken into account (changes marked in blue in the revised version of the manuscript), and a point-by-point answer in given below.

Firstly, the authors need to define what is meant by "1st, 2nd, and 3rd generations of antipsychotics." This will help readers understand the context of the study better.

R: We thank the reviewer for their comment, a paragraph with this information has been added from lines 63 to 77.

Secondly, the mechanism of how nanosystems deliver drugs into the brain through intranasal administration should be more thoroughly explained. This will provide readers with a deeper understanding of the science behind the proposed method of delivery.

R: Thank you for the suggestion, a paragraph on the matter has been added from lines 236 to 243.

Round 2

Reviewer 2 Report

The manuscript was appropriately revised and can be accepted as is.